# Non-Abelian anyon collider

June-Young M. Lee [1] & H.-S. Sim [1] ✉

A collider where particles are injected onto a beam splitter from opposite sides has been used for identifying quantum statistics of identical particles. The collision leads to bunching of the particles for bosons and antibunching for fermions. In recent experiments, a collider was applied to a fractional quantum Hall regime hosting Abelian anyons. The observed negative cross-correlation of electrical currents cannot be understood with fermionic antibunching. Here we predict, based on a conformal field theory and a non-perturbative treatment of non-equilibrium anyon injection, that the collider provides a tool for observation of the braiding statistics of various Abelian and non-Abelian anyons. Its dominant process is not direct collision between injected anyons, contrary to common expectation, but braiding between injected anyons and an anyon excited at the collider. The dependence of the resulting negative cross-correlation on the injection currents distinguishes non-Abelian $\mathrm{SU(2)}_k$ anyons, Ising anyons, and Abelian Laughlin anyons.

Anyons are quasiparticles that are neither fermions nor bosons[1,2]. They exhibit fractional statistics behavior when an anyon winds around another in two dimensions. This is characterized by the overlap, called monodromy, between their states before and after the winding or braiding[3]. While bosons and fermions have the trivial monodromy $M = 1$, Abelian anyons have a complex phase factor $M = e^{-i2\theta}$, where $\theta \neq 0, \pi$ is their position exchange phase. Non-Abelian anyons have a monodromy of $|M| < 1$, as their braiding results in unitary rotation of their state in a degenerate state manifold. The unitary rotation is an element of topological quantum computing[4]. It is expected that along fractional quantum Hall edge channels, there flow anyons such as Abelian Laughlin anyons at filling factor $v = 1/3$, non-Abelian $\mathrm{SU(2)}_{k=2}$ anyons of the anti-Pfaffian state[5,6] or Ising anyons of the particle-hole Pfaffian state at $v = 5/2$[7], and non-Abelian $\mathrm{SU(2)}_{k=3}$ anyons of the anti-Read-Rezayi state at $v = 12/5$[8].

On top of a long time efforts[9–29] on detecting the fractional statistics, there were experimental breakthroughs at $v = 1/3$[30,31]. In a collider experiment[31], two dilute streams of Abelian anyons are injected into a quantum point contact (QPC) that behaves as a collider beam splitter. It shows negative cross-correlations of electrical currents at the output ports of the collider in agreement with a nonequilibrium bosonization theory[26]. It, however, remains unclear which aspect of the Abelian anyon statistics is identified from the experimental result. On one hand, it seems natural to interpret the result as an intermediate between fermionic antibunching and bosonic bunching by the direct collision between injected anyons[26].

On the other hand, a braiding effect was predicted[27,28] in a related setup where Abelian anyons are injected from only one side. The identification is important in pursuing more direct evidence of anyons. It is also intriguing to apply the collider to non-Abelian anyons. There has been no prediction on this issue.

We here develop a theory of a collider encompassing generic Abelian and non-Abelian anyons in fractional quantum Hall systems. We demonstrate that for Abelian and non-Abelian anyons, its dominant process is "time-domain" interference, in which an anyon, excited at the collider $\mathrm{QPC_C}$, braids the injected anyons passing $\mathrm{QPC_C}$ within the interference time window. More anyons are braided as more injected anyons pass. So the cross-correlation depends on the product of the injection current and the monodromy from the braiding, differentiating various anyons. The interference is absent in bosons and fermions, where it corresponds to a trivial vacuum bubble process that does not contribute to observables. Hence the dependence cannot be interpreted as a deviation from fermionic antibunching[32] due to the commonly anticipated direct collision between injected anyons.

## Results

### Non-equilibrium correlator of anyon collider

Figure 1 (a) shows a collider setup. The QPCs of the setup are in the weak backscattering regime that anyon tunneling happens dominantly by the most relevant single type of anyon. Anyons are injected with rate $I_{\mathrm{A/B,inj}}/e^*$ at $\mathrm{QPC_{A/B}}$ by voltage $V_{\mathrm{A/B,inj}}$, and flow to $\mathrm{QPC_C}$ with velocity $v$. The injected anyons are downstream charged anyons or

[1]Department of Physics, Korea Advanced Institute of Science and Technology, Daejeon 34141, Korea. ✉e-mail: hssim@kaist.ac.kr

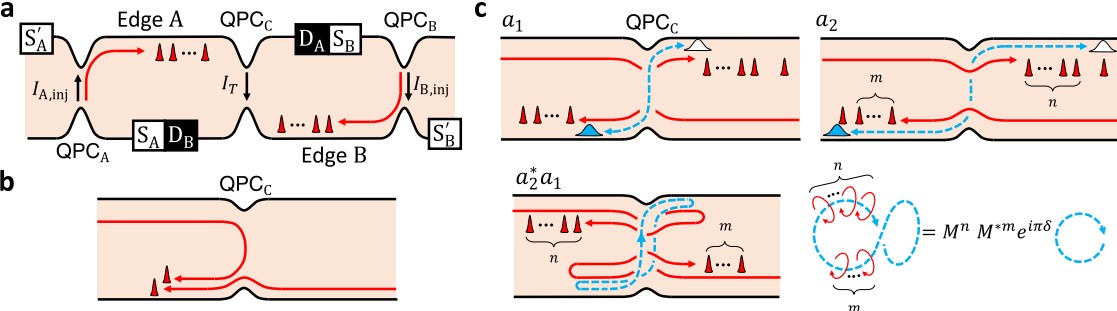

**Fig. 1 | Fractional quantum Hall collider. a** Setup. Anyons are injected to Edge A/B through QPC$_{A/B}$ by voltage $V_{A/B,inj}$ applied to Source S$_{A/B}$, accompanied by current $I_{A/B,inj}$ of charge $e^*$. The injected anyons (red narrow peaks) flow downstream to QPC$_C$ (red trajectories); a corresponding setup for upstream anyons is shown in Fig. 2. The QPCs are in a weak backscattering regime. **b** Conventional collision where an injected anyon collides with another after tunneling at QPC$_C$. **c** Time-domain interference involving $(n, m)$ braiding. Its subprocesses $a_1$ and $a_2$ share the common spatial locations of injected anyons on the Edges. They have tunneling of an additional anyon at QPC$_C$ (blue wide peaks for the anyon, white peaks for its hole counterpart) by thermal or quantum fluctuations, but at different times (blue trajectories). In $a_1$ (resp. $a_2$), the tunneling happens after (resp. before) $n$ and $m$ injected anyons pass QPC$_C$ on Edges A and B. In their interference $a_2^* a_1$, the additional anyon braids the injected anyons, depicted as a blue twisted loop topologically linked with $n$ "counterclockwise" and $m$ "clockwise" red loops. Untying and untwisting the loops give monodromy $M^n(M^*)^m$ and topological spin $e^{i\pi\delta}$.

upstream charge-neutral anyons, and they are not further fractionalized while flowing. Anyon tuneling at QPC$_C$ is described by Hamiltonian $H_T = \mathcal{T}(t) + \mathcal{T}^\dagger(t) = \gamma_C[\psi_B^\dagger(0,t)\psi_A(0,t)]_I$ + h.c.. $\gamma_C$ is the tunneling strength, $\psi_{A/B}^\dagger(x,t)$ creates an anyon on Edge A/B at position $x$ and time $t$, and $[\cdots]_I$ indicates the vacuum fusion channel of the anyon. We consider the dilute injection of $e^*V_{A/B,inj} \gg hI_{A/B,inj}/e^*$ in non-equilibrium with $e^*V_{A/B,inj} \gg k_BT$ at temperature $T$ as in experiments[31], and derive the non-equilibrium correlator of the tunneling operators

$$\langle \mathcal{T}^\dagger(0)\mathcal{T}(t)\rangle_{neq} = e^{-\mathcal{I}t}\langle \mathcal{T}^\dagger(0)\mathcal{T}(t)\rangle_{eq} + \text{subleading terms},$$
$$\mathcal{I} = (1-M)\frac{I_{A,inj}}{e^*} + (1-M^*)\frac{I_{B,inj}}{e^*} \quad (1)$$
$$= \text{Re}\,[1-M]\frac{I_+}{e^*} + i\,\text{Im}[1-M]\frac{I_-}{e^*}$$

for $t > 0$, using the conformal field theory (CFT), Keldysh nonequilibrium theory, and non-perturbative resummation over all perturbation orders of anyon tunneling at QPC$_{A/B}$ (Supplementary Note 1); for $t < 0$, $t \to -t$ and $M \to M'$ are replaced in Eq. (1). $M$ is the monodromy of the injected anyon flowing toward QPC$_C$, which will be discussed later. $\langle\cdots\rangle_{eq}$ is the equilibrium correlator at $V_{A/B,inj} = 0$ and $I_\pm = I_{A,inj} \pm I_{B,inj}$. Equation (1) is valid at $t \gg \hbar/e^*V_{A/B,inj}$.

The current $I_T$ and its zero-frequency noise $\langle \delta I_T^2 \rangle$ at QPC$_C$ are written as $I_T = e^* \int_{-\infty}^{\infty} dt \langle [\mathcal{T}^\dagger(0), \mathcal{T}(t)]\rangle_{neq}$ and $\langle \delta I_T^2 \rangle = e^{*2} \int_{-\infty}^{\infty} dt \langle \{\mathcal{T}^\dagger(0), \mathcal{T}(t)\}\rangle_{neq}$ in the lowest tunneling order $O(|\gamma_C|^2)$ at QPC$_C$, hence, the observables can be directly obtained from Eq. (1). We find

$$I_T = -4e^*|\gamma_C|^2 d_\psi^{-1}\Gamma(1-2\delta)\sin\pi\delta\,\text{Im}\,\mathcal{I}^{2\delta-1} + \text{subleading terms},$$
$$\langle \delta I_T^2 \rangle = 4e^{*2}|\gamma_C|^2 d_\psi^{-1}\Gamma(1-2\delta)\cos\pi\delta\,\text{Re}\,\mathcal{I}^{2\delta-1} + \text{subleading terms}$$
$$(2)$$

at $e^*V_{A/B,inj} \gg hI_{A/B,inj}/e^*$ and zero temperature (see Methods for finite temperature). $d_\psi$ and $\delta$ are the quantum dimension and tunneling exponent of the anyon, and $\Gamma$ is the gamma function. The zero-frequency cross-correlation $\langle \delta I_A \delta I_B \rangle$ of the collider output currents at Detectors D$_A$ and D$_B$ is related with $I_T$ and $\langle \delta I_T^2 \rangle$ (Methods).

## Time-domain interference with anyon braiding

It is remarkable that the observables depend on the product $\mathcal{I}$ of the injection currents $I_{A/B,inj}$ and the monodromy factor $(M-1)$ in Eq. (1).

Its origin, the time-domain interference involving anyon braiding, is identified, using our perturbation approach. We consider an interference event $(n, m)$ between two subprocesses $a_1$ and $a_2$ in a time window $t$. Tunneling of an anyon happens at QPC$_C$ at time $t$ in $a_1$ and at time 0 in $a_2$ [Fig. 1c]. This tunneling occurs not by the voltage $V_{A/B,inj}$ but by thermal excitations, and it is described by the equilibrium correlator $\langle \mathcal{T}^\dagger(0)\mathcal{T}(t)\rangle_{eq}$ in Eq. (1). Within the time window, $n$ anyons on Edge A and $m$ anyons on Edge B pass QPC$_C$ without tunneling. These anyons were injected by $V_{A/B,inj}$. So the interference loop $a_2^* a_1$ in the time axis braids the $n$ anyons on Edge A in a direction and the $m$ anyons on Edge B in the opposite direction, gaining monodromy $M^n(M^*)^m$. The braiding happens with probability $p_A(n,t)p_B(m,t)$ where $p_\alpha(n_\alpha, t) = (\bar{n}^{n_\alpha}/n_\alpha!)e^{-\bar{n}}$ is the Poisson probability distribution for random anyon injections $n_\alpha$ times at QPC$_{\alpha=A,B}$ over time $t$, with an average number $\bar{n}(t,\alpha) = I_{\alpha,inj}t/e^*$. Average of the monodromy over different $(n, m)$'s reproduces the exponential factor in Eq. (1),

$$\exp\left(\frac{I_{A,inj}}{e^*}(M-1)t + \frac{I_{B,inj}}{e^*}(M^*-1)t\right) = \sum_{n,m} p_A(n,t)p_B(m,t)M^n(M^*)^m.$$
$$(3)$$

The validity condition of Eq. (2) with large $V_{A/B,inj}$ is necessary for the braiding; the temporal width $h/(e^*V_{A/B,inj})$ of the injected anyons must be narrower than their separation $e^*/I_{A/B,inj}$ and the window $t \lesssim h/(k_BT)$. The braiding happens even when anyons are injected from only one side, $I_{A,inj} = 0$ or $I_{B,inj} = 0$.

The time-domain interference is distinct from the conventional collision in Fig. 1(b). In the former, the anyon tunneling at QPC$_C$ occurs thermally. In the latter, an anyon injected by the voltage $V_{A/B,inj}$ undergoes tunneling at QPC$_C$. The former dominates over the latter at $e^*V_{A/B,inj} \gg k_BT$ and determines Eq. (2), when the tunneling exponent $\delta$ of QPC$_C$ is smaller than 1 (Supplementary Note 2). This is implied from the voltage dependence $I \sim V^{2\delta-1}$ of QPC tunneling currents in the fractional quantum Hall regime. We note that the factors $\sin\pi\delta$ and $\cos\pi\delta$ in Eq. (2) come from the topological spin or twist factor[3] $e^{i\pi\delta} = e^{i2\pi h_\psi}$ that appears due to operator ordering exchange in the equilibrium correlator $\langle \mathcal{T}^\dagger(0)\mathcal{T}(t)\rangle_{eq}$ for the anyon excited at QPC$_C$, where $h_\psi(=\delta/2)$ is the scaling dimension of the anyon. For Abelian anyons, $e^{i\pi\delta}$ coincides with the exchange phase $e^{i\theta}$.

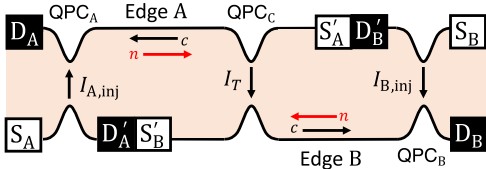

**Fig. 2 | Anyon collider of upstream modes.** It has counter-propagating edge channels, downstream charge modes (black arrows, label $c$) and upstream neutral modes (red arrows, label $n$). In this setup, the injection current $I_{A/B,inj}$ at QPC$_{A/B}$ results in the flow of upstream modes from QPC$_{A/B}$ to QPC$_C$ on Edge A/B. The locations of the charge sources (S$_{A/B}$,S$'_{A/B}$) and detectors (D$_{A/B}$,D$'_{A/B}$) are different from Fig. 1a.

## Fano factor

The dependence of the observables on the product $\mathcal{I}$ in Eq. (2) offers possibility of observing anyon braiding. The Fano factor $P_-(I_-/I_+) \equiv \dfrac{\langle \delta I_A \delta I_B \rangle}{e^* I_+ \frac{\partial I_T}{\partial I_-}|_{I_-=0}}$ introduced in ref. 26 is useful. When $I_{A,inj} = I_{B,inj}$, we find

$$P_-(0) = 1 - \frac{\mathrm{Re}[1-M]}{\mathrm{Im}[1-M]} \frac{\cot \pi \delta}{1-2\delta} \quad (4)$$

at zero temperature. For Abelian anyons, $M = e^{-2i\theta}$, then Eq. (4) becomes identical to the expression that was found in ref. 26 but without recognition of the braiding. The dependence of $P_-$ on $I_-/I_+$ was observed at $\nu = 1/3$[31]. Our time-domain interference implies that the observation is an evidence of Abelian anyon braiding.

## Application to non-Abelian anyons

Our findings are equally applicable to non-Abelian anyons. On the most promising non-Abelian states such as anti-Pfaffian[5,6] and particle-hole symmetric Pfaffian state at $\nu = 5/2$[7], or anti-Read-Rezayi state at $\nu = 12/5$[8], the tunneling at QPCs generates downstream Abelian anyons and upstream non-Abelian anyons together. Hence, one can inject the former or latter selectively into QPC$_C$, to observe its braiding. We focus on the case that upstream non-Abelian anyons flow from QPC$_{A/B}$ to QPC$_C$ on Edge A/B (Fig. 2). In this case, anyon tunneling happens at QPC$_{A/B}$ with rate $I_{A/B,inj}/e^*$. The tunneling results in downstream current $I_{A/B,inj}$ of Abelian anyons of charge $e^*$ flowing toward D$_{A/B}$, and upstream charge-neutral mode of the non-Abelian anyons that propagate toward QPC$_C$ and experience braiding with another non-Abelian anyon excited at QPC$_C$ as in the collider at $\nu = 1/3$. Although the non-Abelian anyon excited at QPC$_C$ is charge neutral, the excitation is always accompanied by tunneling of a charged Abelian anyon, giving rise to charge currents detected at D$_A$ or D$_B$. Hence the braiding information can be read out from $\langle \delta I_A \delta I_B \rangle$. Side effects by back flows from QPC$_C$ to QPC$_{A/B}$ are negligible in our parameter regime (Supplementary Note 4), and Eqs. (1), (2), and (4) are also valid for the non-Abelian anyons. In the equations, $\delta$ is the tunneling exponent of a composite of the charged Abelian anyon and the neutral non-Abelian anyon that together tunnel at QPC$_C$, while $M$ is the monodromy of only the non-Abelian anyon since the braiding happens between the non-Abelian anyon and other injected non-Abelian anyons.

The non-Abelian anyons at $\nu = 5/2$ and 12/5 have Im[$M$] = 0 (see their monodromy in Fig. 3). As a notable result, the time-domain interference contributes to the current $I_T$ destructively [see Eqs. (1) and (2)], and Fano factor $P_-(0)$ diverges; the divergence is regularized, $P_- \sim O\left( \left( \frac{(e^*)^2}{h} \frac{V_{A/B,inj}}{I_{A/B,inj}} \right)^{h_a} \right)$, by the subleading terms in Eq. (2), where $h_a$ is the scaling dimension of a fusion channel different from the vacuum (Supplementary Note 2). For quantitative comparison among anyons, we suggest another Fano factor $P_{ref}(I_-/I_+) \equiv (e^* e/h) \langle \delta I_A \delta I_B \rangle / (I_+ \partial I_T/\partial V_{ref}|_{I_-=0, V_{ref}=0})$ defined with a small reference voltage $V_{ref}$

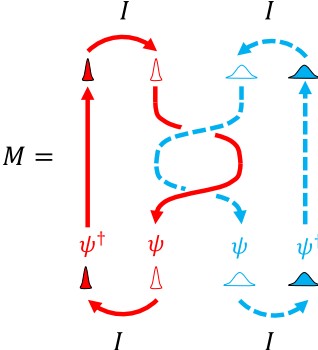

**Fig. 3 | Monodromy $M$ for non-Abelian anyons.** Two particle-hole pairs of $\psi$ anyons are initially split from the vacuum ($I$). After the braiding, they fuse into the vacuum. The monodromy is the amplitude of this process. The red and blue loops correspond to anyons that tunnel at QPC$_{A/B}$ and QPC$_C$, respectively [See the loops of the same colors in Fig. 1c]. Untying the topological link between the loops amounts to the monodromy $M$ (or $M'$ depending on the direction of the loops).

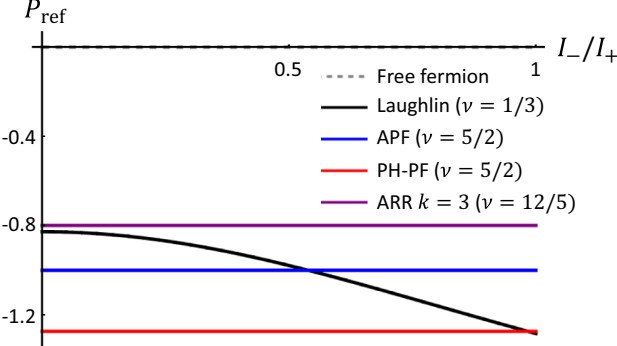

**Fig. 4 | Dependence of Fano factor $P_{ref}$ on $I_-/I_+$ for various anyons.** The Fano factors are shown for free fermions (gray dashed), Laughlin anyons at $\nu = 1/3$ (black), anti-Pfaffian state at $\nu = 5/2$ (APf, blue), particle-hole Pfaffian state at $\nu = 5/2$ (PH-Pf, red), and anti-Read-Rezayi state at $\nu = 12/5$ (ARR, purple). At any value of $I_-/I_+$, $P_{ref} = -1$ for the anti-Pfaffian state, $P_{ref} = -\pi/4$ for the particle-hole Pfaffian state, and $P_{ref} = -5\sqrt{250 - 110\sqrt{5}}/4\pi \simeq -0.8$ for the anti-Read-Rezayi state. The behaviors of the non-Abelian anyons are distinguished from free fermions with $P_{ref} = 0$ and the Abelian anyons.

applied to Source S$'_A$ and voltage shift $V_{A,inj} \to V_{A,inj} + V_{ref}$ at Source S$_A$ (the voltage across QPC$_A$ remains as $V_{A,inj}$). We find $P_{ref}(I_-/I_+) = P_-(I_-/I_+) \mathrm{Im}[1-M] e/(2\pi e^*)$. When $I_{A,inj} = I_{B,inj}$,

$$P_{ref}(0) = \frac{\mathrm{Im}[1-M]}{2\pi e^*/e} - \frac{\mathrm{Re}[1-M]}{2\pi e^*/e} \frac{\cot \pi \delta}{1-2\delta}. \quad (5)$$

$P_{ref}$ is notably independent of $I_-/I_+$ for the non-Abelian anyons having Im[$M$] = 0. In Fig. 4, the behavior of $P_{ref}$ distinguishes various anyons. $P_{ref}$ also differs between the anti-Pfaffian state and the particle-hole Pfaffian state at $\nu = 5/2$; the states have $M = 0$ in common but different $\delta$.

$P_{ref}$ is experimentally measurable (Methods). It is also possible to gain monodromy information from $I_T$ without measuring $\langle \delta I_A \delta I_B \rangle$ (Methods).

We note that there are some non-Abelian states, e.g., the Pfaffian state at $\nu = 5/2$[33], in which a Abelian charge mode and a non-Abelian neutral mode co-propagate along edges. In those cases, the neutral mode propagates typically slower than the charge mode, and Eq. (1) is not directly applicable. The multiplicative factor of Eq. (1) is modified

non-universally, depending on the velocities of the modes and the distances between the QPCs.

## Discussion

We compare the time-domain interference with a Fabry–Perot interference[14,17–22,25,30]. In the latter, an anyon moving around the edge of a Fabry–Perot cavity braid localized bulk anyons inside the cavity. It is detected in the linear response of the interference current, with changing the number of the localized anyons by a gate voltage. It corresponds to the interference of free fermions where the braiding is trivial. By contrast, in the former, braiding happens between anyons on one-dimensional edges, as the time ordering provides an extra dimension for braiding. It is detected in the non-equilibrium response, with changing the number of injected anyons by $I_{A/B,inj}$.

The time-domain interference is absent in free fermions of $M = 1$. For them, the exponential factor in Eq. (1) becomes the trivial value 1, and the leading contributions in Eq. (2) vanish. It is because the topological link between the blue and red loops in Fig. 1c becomes trivial. The blue loop is completely independent of the red loops, constituting a disconnected Feynman diagram (a vacuum bubble) in the perturbation theory. This diagram cannot contribute to observables, as its contribution $M$ to the interference is exactly canceled, $M - 1 = 0$, by the trivial value 1 from a partner disconnected diagram, according to the linked cluster theorem. By contrast, in Abelian and non-Abelian anyons, the cancellation is only partial, $M - 1 \neq 0$. We notice, in every perturbation order, the pairwise appearance of a braiding diagram and its partner disconnected diagram resulting in the factor $M - 1$ (Supplementary Note 1). This explains $M - 1$ in Eqs. (1) and (2). As the time-domain interference has no counterpart in free fermions, the result in Fig. 4 cannot be interpreted as a deviation from fermionic antibunching due to the direct collision.

Our computation methods, results, and interpretations are based on the bulk-edge correspondence of topological order. The edge of a topological order is described by a certain CFT, whose primary fields correspond to the anyons of the topological order. The wavefunction of "bulk" anyons localized in the bulk of the topological order can be written as the correlator of the primary fields[4]. The braiding statistics of the anyons is encoded in the duality matrices of the corresponding conformal block. Hence, one can obtain information about the braiding statistics among "edge" anyons propagating along the edge, using the anyon collider or similar setups, without involving bulk anyons. Meanwhile, however, the edge anyons are not protected by the energy gap of the topological order, and the structure of the CFT can be altered by various mechanisms[34,35] such as decoherence and edge reconstruction. Then the monodromy $M$ and the topological spin $\delta$ can have values different from those of the topological order.

We discuss experimental observability. To observe the Fano factor $P_-$, phase coherence of Edge A/B is required near QPC$_C$ over a distance longer than the thermal length $\hbar v/(k_B T)$, where $v$ is the anyon velocity. Edge reconstruction[34] needs to be avoided over the distance, as it modifies $M$ and $\delta$. It is also required that QPC$_C$ follows the power law $I \sim V^{2\delta-1}$ in an energy window which covers the voltages $e^* V_{A/B,inj}$ and temperature $k_B T$. The requirements may be achieved in experiments[31]. When the energy window also includes the small voltage $e^* V_{ref}$, the Fano factor $P_{ref}$ can be measured. Note that the bulk-edge coupling of non-Abelian anyons[21–23] and Coulomb interaction[24,25] of a Fabry–Perot cavity may be irrelevant in the collider.

It is interesting that the braiding effect appears and dominates the observables in the collider. It differs from the conventional collision, and provides a tool for identifying the braiding of various Abelian and non-Abelian anyons. Our finding implies that recent collider experiments[31], in fact, provide a signature of Abelian anyon braiding, rather than the (anti)bunching effects commonly recognized by the

community. Our theory is applicable to other topological orders, as it is based on the generic CFT. The time-domain interference will be useful for identifying fractional statistics in systems having no topological order[36,37] and for the engineering mobile anyons with tuning edge channels by electrical gates.

## Methods
### Tunneling current and noise

We provide the expression of the electrical current $I_T$ and its zero-frequency noise $\langle \delta I_T^2 \rangle$ at QPC$_C$ at temperature $k_B T \ll e^* V_{A/B,inj}$ and $h I_{A/B,inj}/e^* \ll e^* V_{A/B,inj}$,

$$I_T = e^* \int_{-\infty}^{\infty} dt \langle [\mathcal{T}^\dagger(0), \mathcal{T}(t)] \rangle_{neq} = -C e^* (k_B T)^{2\delta-1} \text{Im} \left[ \frac{\Gamma(\mathcal{I}/2\pi k_B T + \delta)}{\Gamma(\mathcal{I}/2\pi k_B T + 1 - \delta)} \right],$$

$$\langle \delta I_T^2 \rangle = e^{*2} \int_{-\infty}^{\infty} dt \langle \{\mathcal{T}^\dagger(0), \mathcal{T}(t)\} \rangle_{neq} = \frac{C e^{*2}}{\tan \pi \delta} (k_B T)^{2\delta-1} \text{Re} \left[ \frac{\Gamma(\mathcal{I}/2\pi k_B T + \delta)}{\Gamma(\mathcal{I}/2\pi k_B T + 1 - \delta)} \right],$$

(6)

where $C = 4(2\pi)^{2\delta-1} |\gamma_C|^2 \Gamma(1 - 2\delta) \sin \pi\delta / d_\psi$. This is the generalization of the zero-temperature result for Abelian anyons in ref. 26 to Abelian or non-Abelian anyons at finite temperature.

### Cross-correlation

The cross-correlation $\langle \delta I_A \delta I_B \rangle$ is related with $I_T$ and $\langle \delta I_T^2 \rangle$. Using the charge conservation, we derive the zero-temperature relations of

$$\langle \delta I_A \delta I_B \rangle = -\langle \delta I_T^2 \rangle + \langle \delta I_{A,inj} \delta I_T \rangle - \langle \delta I_{B,inj} \delta I_T \rangle + \langle \delta I_{A,inj} \delta I_{B,inj} \rangle,$$

$$\langle \delta I_{A(B),inj} \delta I_T \rangle = e^* I_{A(B),inj} \frac{\partial I_T}{\partial I_{A(B),inj}}.$$

(7)

The latter relation is valid when Im[$M$]$\neq 0$. In Supplementary Note 3, the derivation of the relations, $\langle \delta I_{A/B,inj} \delta I_T \rangle$, and $\langle \delta I_{A,inj} \delta I_{B,inj} \rangle$ is found, and the case of Im[$M$] = 0 is discussed.

### Symmetric injection

In the nearly symmetric injection case of $I_{A,inj} \simeq I_{B,inj}$ or $I_+ \gg I_-$, the zero-temperature expressions of $I_T$ and $\langle \delta I_T^2 \rangle$ at QPC$_C$ in Eq. (2) are simplified as

$$I_T \simeq \frac{4|\gamma_C|^2 e^*}{d_\psi \csc \pi\delta} (1 - 2\delta) \Gamma(1 - 2\delta) \left( \text{Re}[1 - M] \frac{I_+}{e^*} \right)^{2\delta-2} \left( \frac{e^* V_{ref}}{\hbar} - \text{Im}[M] \frac{I_-}{e^*} \right),$$

$$\langle \delta I_T^2 \rangle \simeq \frac{4|\gamma_C|^2 e^{*2}}{d_\psi \sec \pi\delta} \Gamma(1 - 2\delta) \left( \text{Re}[1 - M] \frac{I_+}{e^*} \right)^{2\delta-1}.$$

(8)

Here we consider the situation where the voltage $V_{A,inj} + V_{ref}$ is applied at Source S$_A$, $V_{B,inj}$ is at Source S$_B$, and a very small voltage $V_{ref}$ is at Source S$'_A$. In this situation, the voltage cross QPC$_A$ remains as $V_{A,inj}$. The effect of $V_{ref}$ does not modify Eq. (2) except the replacement of $\mathcal{I} \to \mathcal{I} = \text{Re}[1 - M] \frac{I_+}{e^*} + i \text{Im}[1 - M] \frac{I_-}{e^*} + i \frac{e^*}{\hbar} V_{ref}$. $V_{ref}$ decouples from the monodromy factor $(1 - M)$ in $\mathcal{I}$, as it does not cause any braiding.

### Properties of non-Abelian anyons

We briefly introduce the anti-Read-Rezayi (ARR) state at level-$k$, a promising candidate hosting non-Abelian anyonic excitations[8]. It has been expected that it is the ground state at $\nu = 2 + \frac{2}{k+2}$. In particular, the ARR states of level 2 and of level 3 correspond to the anti-Pfaffian state at $\nu = 5/2$ and the ARR state at $\nu = 12/5$, respectively. The edge-channel structure of the level-$k$ ARR state is decomposed, as a result of random inter-edge tunneling[5,6,8], into downstream charge modes, described by the free boson CFT, and an upstream neutral mode, described by the SU(2)$_k$ Wess-Zumino-Witten CFT. There are two types of quasiparticles with the smallest scaling dimension of $h_\psi = 1/(k+2)$ and hence the smallest tunneling exponent $\delta = 2h_\psi = 2/(k+2)$ for ideal edges, one carrying only charge $e^* = 2e/(k+2)$, and the other carrying $e^* = e/(k+2)$

and the neutral part $j = 1/2$ in the context of the $SU(2)_k$ anyons. As the bare tunneling strength of the former at a QPC is expected to be much smaller than the latter, we assume that tunneling at the QPCs is dominated by the latter having non-Abelian anyons in the neutral part. The monodromy of the non-Abelian anyons is $M = \frac{\cos(2\pi/(k+2))}{\cos(\pi/(k+2))}$.

We also consider the particle-hole symmetric Pfaffian state, another competitive ground state candidate of $v = 5/2$[29]. Its edge structure is similar to the anti-Pfaffian state, except that the neutral mode is described by the Ising CFT, and charge $e/4$ quasiparticle contains the non-Abelian anyonic $\sigma$ primary field with a scaling dimension of $1/8$[7]. The monodromy of the non-Abelian anyon is $M = 0$.

### Differential conductances

We suggest how $\partial I_T/\partial I_-$ and $\partial I_T/\partial V_{ref}$, hence, the Fano factors $P_-$ and $P_{ref}$, can be obtained from standard lock-in measurements. First, to obtain $\partial I_T/\partial I_-$, one applies a small AC voltage to Source $S'_A$ in the presence of the voltages $V_{A/B,inj}$ at $QPC_{A/B}$, and measures the AC current at Detector $D_B$. Then one gets the differential conductance of

$$\frac{dI^{(1)}_{D_B}}{dV} = \frac{dI_T}{dV} = \frac{\partial I_T}{\partial I_{A,inj}}\frac{\partial I_{A,inj}}{\partial V_{A,inj}} = G T_A \frac{\partial I_T}{\partial I_{A,inj}}. \quad (9)$$

$G$ is the conductance quantum $e^*e/h$. $T_A \equiv G^{-1}\partial I_{A,inj}/\partial V_{A,inj}$ is the transmission probability at $QPC_A$, and it can be measured by another lock-in measurement. From this, one can obtain $\partial I_T/\partial I_{A,inj}$. In the limit of $I_- = 0$, $\partial I_T/\partial I_-$ is identical to $\partial I_T/\partial I_{A,inj}$. At nonzero $I_-$, one has a similar measurement for $\partial I_T/\partial I_{B,inj}$, and obtains $\partial I_T/\partial I_- = (\partial I_T/\partial I_{A,inj} - \partial I_T/\partial I_{B,inj})/2$.

Next, to obtain $\partial I_T/\partial V_{ref}$, one applies a small AC voltage to Source $S'_A$ in the presence of the voltages $V_{A/B,inj}$ at $QPC_{A/B}$, and measures the AC current at Detector $D_B$. Then one gets the differential conductance of

$$\frac{dI^{(2)}_{D_B}}{dV} = \frac{dI_T}{dV} = -\frac{\partial I_T}{\partial I_{A,inj}}\frac{\partial I_{A,inj}}{\partial V_{A,inj}} + \frac{\partial I_T}{\partial V_{ref}} = -G T_A \frac{\partial I_T}{\partial I_{A,inj}} + \frac{\partial I_T}{\partial V_{ref}}. \quad (10)$$

Combining $dI^{(1)}_{D_B}/dV$ and $dI^{(2)}_{D_B}/dV$, one can obtain $\partial I_T/\partial V_{ref}$. Note that in the equality in Eq. (10), $I_{A,inj}$ and $V_{ref}$ are treated as independent variables. It is because we consider the situation of the voltage $V_{A,inj} + V_{ref}$ applied at Source $S_A$, $V_{B,inj}$ at Source $S_B$, and a very small voltage $V_{ref}$ at Source $S'_A$; in this situation, the voltage across the $QPC_A$ (hence $I_{A,inj}$) is independent of $V_{ref}$.

It is possible to gain monodromy information from the differential conductances without measuring the cross-correlation, since the time-domain interference involving the braiding affects the tunneling current $I_T$. From Eqs. (2) and (8), we find that the ratio of the differential conductances depends only on the fractional charge and $Im[M]$,

$$\frac{\partial I_T/\partial I_-|_{V_{ref}=0}}{\partial I_T/\partial V_{ref}|_{V_{ref}=0}} = \frac{\hbar}{(e^*)^2} Im[1-M]. \quad (11)$$

Interestingly, this ratio is independent of $I_+$ and $I_-$. For those non-Abelian anyons having $Im[M] = 0$, this ratio shows a vanishingly small value of $O\left(\left(\frac{\hbar}{(e^*)^2}\frac{I_{A/B,inj}}{V_{A/B,inj}}\right)^{h_a}\right)$. The ratio can be measured when $QPC_C$ follows the power law $I \sim V^{2\delta-1}$ in an energy window which covers the voltages $e^*V_{A/B,inj}$, temperature $k_BT$, and small voltage $e^*V_{ref}$.

## Data availability

All the calculation details are provided in Supplementary Information.

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

## Acknowledgements

We thank Anne Anthore, Hyung Kook Choi, Gwendal Feve, Christian Glattli, Donghoon Kim, Christophe Mora, and Frederic Pierre for useful discussions. H.-S.S. acknowledges support from Korea NRF, the SRC Center for Quantum Coherence in Condensed Matter (Grant No. 2016R1A5A1008184). J.-Y.M.L. acknowledges support from Korea NRF, NRF-2019-Global Ph.D. fellowship.

## Author contributions

J.-Y.M.L. performed the detailed calculations, analyzed the results, and wrote the paper. H.-S.S. supervised the project, analyzed the results, and wrote the paper.

## Competing interests

The authors declare no competing interests.
