## [Peer Review File · Nature Communications]

Non-Abelian Anyon ColliderREVIEWER COMMENTS

Reviewer #1 (Remarks to the Author):

Recently, a striking anyon collider experiment has been performed. It produced evidence of Abelian anyonic statistics at the simplest filling factor $1/3$ in the fractional quantum Hall effect. This evidence is somewhat indirect in the sense that it depends on aspects of edge physics. Nevertheless, it was a very important experimental achievement, especially, given that even the most direct interferometry experiments are inevitably sensitive to various details of the sample and so an ideally direct experiment is arguably impossible.

This manuscript extends theoretical analysis of anyon colliders to a general anyonic statistics, most importantly, non-Abelian statistics. The non-Abelian extension is important. It is a challenging technical task and the authors should be congratulated on successfully solving the problem. In my view, the non-Abelian side of the paper should be emphasized as its main achievement.

The authors also provide an alternative interpretation of the collider experiment, but this is more of a language exercise. For example, the role of edge reconstruction makes the experiment inevitably somewhat indirect. Moreover, the calculation depends on a critical exponent associated with anyons. Such exponents are routinely non-universal experimentally. I object the authors' claim that the collider experiment is more direct than the other approaches.

The above point does not subtract from the authors' technical achievement. I believe that the paper can be published in Nature Communications after a modest revision.

Reviewer #2 (Remarks to the Author):

The manuscript "Non-Abelian anyon collider" by Lee and Sim, presents a theoretical study of the collider geometry in the fractional quantum effect for investigating the fractional statistics of anyons. The case of abelian anyons has already been theoretically and experimentally investigated in Refs [2] and [3]. The manuscript deals with the non-abelian case, for which predictions were missing so far, and which is relevant for the filling factors $5/2$ and $12/5$ for example.

I think this work is important for the two following reasons. Firstly, it provides an important step forward in the understanding of collision experiments and on the properties of anyons that are probed in collision experiments (in the abelian and non-abelian cases). Fractional statistics show up as negative values of cross-correlations that can be understood as resulting from the bunching of particles at the output of the collider. The authors show that this behavior can be explicitly related to time domain interferences involving anyon braiding. It means this time domain braiding mechanisms, which is dominant for anyons, is absent in the case of fermions or bosons such that current cross-correlations directly probe the monodromy $M = \exp(-2i \theta)$ where θ is the exchange phase via the factor $1-M$. This understanding of the anyon collider allows then to extend its range of application to the non-abelian case where the monodromy is not a simple phase but satisfies $|M| < 1$. This is the second achievement of the paper: to provide quantitative predictions for the value of the cross-correlations in the collider geometry in the non-abelian case. The authors show that colliders are able to discriminate between abelian and non-abelian statistics and even between different non-abelian topological orders.

The applicability of the collider geometry to the non-abelian case was absolutely not settled before this work. In particular, in the non-abelian topological case, anyons fractionalize into an abelian charged part and a neutral non-abelian one. Probing non-abelian statistics in the collider geometry was thus an open question for two main reasons. As discussed above, the fact that the collider probes the monodromy was not fully understood. The second difficulty was related to the neutral nature of the

non-abelian part of anyons. It was thus unclear how current noise measurements could probe neutral excitations. In this work, the authors show that, as anyon emission implies the simultaneous generation of the charged abelian part and the neutral non-abelian one, one can read the braiding processes of the neutral excitations from the current noise generated on the charged part.

For all these reasons, I think this work represents an important step forward in the state of the art of the field and I thus recommend the paper for publication in Nature Communications. I have however a few remarks that could improve the readability of the manuscript.

My main comment regards the differences between the two geometries where the charged abelian part and the neutral abelian one are co-propagating or on the contrary are counterpropagating. In most of the realistic cases (anti-pfaffian and particle-hole pfaffian) we are in the second situation with a forward propagating charge mode and a backward propagating neutral mode. In this case the authors should explain more explicitly in the text the meaning of Eq. (1):

-what are the currents $I_{A,inj}$ and $I_{B,inj}$? I suppose these are the charged currents that do not propagate towards QPCc but rather towards the ohmics DA and DB (Fig S3).

-what is the charge e^* , I assume it is the again the charge of the charged mode?

-does the monodromy in Eq. (1) take only into account the neutral non-abelian part? Does that mean that the result would be different in the case where both the charge and neutral mode co-propagate, in which case the monodromy would also include the braiding of the abelian charge mode?

All these questions should be discussed in the main text of the manuscript.

I also have a related questions, which is even more puzzling for me, regarding the differential conductance, eq. (9). As far as I understand, the main text discusses the case where the charge mode propagates towards QPCc, in which case I understand how to measure the tunneling current resulting from the injection of currents I_A and I_B . However, I do not understand how one can define the differential conductance in the other regime where the neutral mode is propagating towards QPCc. It seems to me that the tunneling current is always zero as no net current propagates towards QPCc (currents I_A and I_B propagate towards DA and DB on Fig S3). I would thus conclude that, in this case, I_T is always zero and independent of I_A and V_{ref} and I do not understand how Perf is defined in this case?

Finally, Fig 2 looks like a useful of the monodromy but I find it hard to connect it to the Fig1.c, that is, to understand how the diagram in Fig.2 enters in the description of the collision experiment. Is it possible to make this connection more explicit?

Reviewer's overall comment: "Recently, a striking anyon collider experiment has been performed. It produced evidence of Abelian anyonic statistics at the simplest filling factor $1/3$ in the fractional quantum Hall effect. This evidence is somewhat indirect in the sense that it depends on aspects of edge physics. Nevertheless, it was a very important experimental achievement, especially, given that even the most direct interferometry experiments are inevitably sensitive to various details of the sample and so an ideally direct experiment is arguably impossible.

This manuscript extends theoretical analysis of anyon colliders to a general anyonic statistics, most importantly, non-Abelian statistics. The non-Abelian extension is important. It is a challenging technical task and the authors should be congratulated on successfully solving the problem. In my view, the non-Abelian side of the paper should be emphasized as its main achievement.

... (the comment here is separately shown below in Reviewer's comment-1)...

I believe that the paper can be published in Nature Communications after a modest revision."

Response: We greatly thank Reviewer 1 for pointing out the importance of our work and regarding our work as successful solving of a challenging technical task, and also for the very positive recommendation. We revise our manuscript as below, taking the comments into account.

Reviewer's comment-1: "The authors also provide an alternative interpretation of the collider experiment, but this is more of a language exercise. For example, the role of edge reconstruction makes the experiment inevitably somewhat indirect. Moreover, the calculation depends on a critical exponent associated with anyons. Such exponents are routinely non-universal experimentally. I object the authors' claim that the collider experiment is more direct than the other approaches.

The above point does not subtract from the authors' technical achievement."

Response: We thank Reviewer 1 for the comment.

Our manuscript provides understanding about the properties (braiding and monodromy) of anyons that can be probed in collider experiments, in addition to the extension to non-Abelian anyons. We would like to emphasize that our interpretation based on anyon braiding is now being accepted by the communities including those theoretically and experimentally working on the colliders.

About the non-universal effects, we agree with Reviewer 1 that non-universal effects, such as edge reconstruction and non-ideal QPCs, affect experiments, as they can modify the critical exponent or the monodromy on which the collider signals rely. As Reviewer 1 mentioned, most approaches, including the Fabry-Perot interferometer and the collider, utilize edge channels and QPCs, so it is a challenge to pursue an ideal experiment for detecting anyon braiding. At the same time, experiments are going to more relevant parameter regimes with more avoiding those non-universal effects. We heard that more than three leading experimental groups are working on anyon colliders. We hope that they find important results.

In our manuscript, we used the expression "direct signal of anyon braiding" in the sense that the collider signals (the results by Bartolomei et al., and those reproduced by other experimental groups) cannot be explained by the fermionic statistics and conventional collision processes, but require our time-domain interference process involving anyon braiding. From the comment of Reviewer 1, we now feel that we need to be careful for using the expression. In the revised manuscript, we omit the word "direct" in the expression.

Moreover, we add the following sentences in the section "Discussions", to connect our results with the bulk topological order and discuss the limitation of the connection due to non-universal effects: "Our computation methods, results, and interpretations are based on the bulk-edge correspondence of a topological order. The edge of a topological order is described by a certain CFT, whose primary fields correspond to the anyons of

the topological order. The wavefunction of "bulk" anyons localized in the bulk of the topological order can be written as the correlator of the primary fields. The braiding statistics of the anyons is encoded in the duality matrices of the corresponding conformal block. Hence, one can obtain the information about the braiding statistics among "edge" anyons propagating along the edge, using the anyon collider or similar setups, without involving bulk anyons. Meanwhile, however, the edge anyons are not protected by the energy gap of the topological order, and the structure of the CFT can be altered by various mechanisms such as decoherence and edge reconstruction. Then the monodromy M and the topological spin δ can have values different from those of the topological order."

Thanks to the comments, we improve our manuscript. We would like to thank Reviewer 1 very much.

Reply to the report of Reviewer 2 (Non-Abelian Anyon Collider by June-Young M. Lee and H.-S. Sim)

Reviewer's overall comment: "The manuscript "Non-Abelian anyon collider" by Lee and Sim, presents a theoretical study of the collider geometry in the fractional quantum effect for investigating the fractional statistics of anyons. The case of abelian anyons has already been theoretically and experimentally investigated in Refs [2] and [3]. The manuscript deals with the non-abelian case, for which predictions were missing so far, and which is relevant for the filling factors $5/2$ and $12/5$ for example.

I think this work is important for the two following reasons. Firstly, it provides an important step forward in the understanding of collision experiments and on the properties of anyons that are probed in collision experiments (in the abelian and non-abelian cases). Fractional statistics show up as negative values of cross-correlations that can be understood as resulting from the bunching of particles at the output of the collider. The authors show that this behavior can be explicitly related to time domain interferences involving anyon braiding. It means this time domain braiding mechanisms, which is dominant for anyons, is absent in the case of fermions or bosons such that current cross-correlations directly probe the monodromy $M = \exp(-2i\theta)$ where θ is the exchange phase via the factor $1-M$. This understanding of the anyon collider allows then to extend its range of application to the non-abelian case where the monodromy is not a simple phase but satisfies $|M| < 1$.

This is the second achievement of the paper: to provide quantitative predictions for the value of the cross-correlations in the collider geometry in the non-abelian case. The authors show that colliders are able to discriminate between abelian and non-abelian statistics and even between different non-abelian topological orders.

The applicability of the collider geometry to the non-abelian case was absolutely not settled before this work. In particular, in the non-abelian topological case, anyons fractionalize into an abelian charged part and a neutral non-abelian one. Probing non-abelian statistics in the collider geometry was thus an open question for two main reasons. As discussed above, the fact that the collider probes the monodromy was not fully understood. The second difficulty was related to the neutral nature of the non-abelian part of anyons. It was thus unclear how current noise measurements could probe neutral excitations. In this work, the authors show that, as anyon emission implies the simultaneous generation of the charged abelian part and the neutral non-abelian one, one can read the braiding processes of the neutral excitations from the current noise generated on the charged part.

For all these reasons, I think this work represents an important step forward in the state of the art of the field and I thus recommend the paper for publication in Nature Communications. I have however a few remarks that could improve the readability of the manuscript."

Response: We greatly appreciate Reviewer 2 for carefully reading the manuscript and elaborating the importance of our work point by point. We also thank the very positive recommendation. We revise our manuscript as below, taking the comments of Reviewer 2 into account.

Reviewer's comment-1: "My main comment regards the differences between the two geometries where the charged abelian part and the neutral abelian one are co-propagating or on the contrary are counterpropagating. In most of the realistic cases (anti-pfaffian and particle-hole pfaffian) we are in the second situation with a forward propagating charge mode and a backward propagating neutral mode. In this case the authors should explain more explicitly in the text the meaning of Eq. (1):

-what are the currents $I_{A,inj}$ and $I_{B,inj}$? I suppose these are the charged currents that do not propagate towards QPCc but rather towards the ohmics DA and DB (Fig S3).

-what is the charge e^* , I assume it is the again the charge of the charged mode?

-does the monodromy in Eq. (1) take only into account the neutral non-abelian part? Does that mean that the result would be different in the case where both the charge and neutral mode co-propagate, in which case the monodromy would also include the braiding of the abelian charge mode?

All these questions should be discussed in the main text of the manuscript."

Response: We thank Reviewer 2 for the questions and suggestions. They are indeed useful for improving the readability of our manuscript. The reviewer's understandings are all correct. $I_{A,inj}$, $I_{B,inj}$ are the charge currents towards the ohmics DA and DB. e^* is the charge of the charge mode that tunnels together with the neutral mode. And the monodromy takes only the neutral non-Abelian part into account, when the charged part and the neutral part are counterpropagating and only the neutral part is injected to the collider QPC.

To clarify the points and improve the readability, we add a figure (Fig. 2) for the setup for injection of upstream neutral mode of non-Abelian anyons to the collider QPC, which was Fig. S3 of the previous version of the Supplementary Materials. We also include the following sentences in the section "Application to non-Abelian anyons": "We focus on the case that upstream non-Abelian anyons flow from QPC_A/B to QPC_C on Edge A/B (Fig. 2). In this case, anyon tunneling happens at QPC_A/B with rate $I_{A/B,inj} / e^*$. The tunneling results in downstream current $I_{A/B,inj}$ of Abelian anyons of charge e^* flowing toward D_A/B, and upstream charge neutral mode of the non-Abelian anyons that propagate toward QPC_C and experience braiding with another non-Abelian anyon excited at QPC_C as in the collider at $\nu=1/3$. Although the non-Abelian anyon excited at QPC_C is charge neutral, the excitation is always accompanied by tunneling of a charged Abelian anyon, giving rise to charge currents detected at D_A or D_B. Hence the braiding information can be read out from $\langle \delta I_A \delta I_B \rangle$ " and "In the equations, δ is the tunneling exponent of a composite of the charged Abelian anyon and the neutral non-Abelian anyon that together tunnel at QPC_C, while M is the monodromy of only the non-Abelian anyon since the braiding happens between the non-Abelian anyon and other injected non-Abelian anyons."

In the case that the charge mode and the neutral mode co-propagate along edges, the braiding depends on their velocities. When their velocities are same, Eq. (1) is directly applicable and the monodromy of Eq. (1) equals the product of the charge-mode monodromy and the neutral-mode monodromy, as the reviewer expected. However, the neutral mode is expected to be slower than the charge mode. Then the injected composite of a charged anyon and a neutral anyon can be fractionalized while flowing, and the multiplicative factor in Eq. (1) has a complicated form that depends on the velocities and the distance between QPC_A/B and QPC_C.

To discuss this, we add the followings in the section "Application to non-Abelian anyons": "We note that there are some non-Abelian states, e.g., the Pfaffian state at $\nu = 5/2$, in which a Abelian charge mode and a non-Abelian neutral mode co-propagate along edges. In those cases, the neutral mode propagates typically slower than the charge mode, and Eq. (1) is not directly applicable. The multiplicative factor of Eq. (1) is modified non-universally, depending on the velocities of the modes and the distances between the QPCs."

And to more clarify the validity condition of Eq. (1), we mention "The injected anyons are downstream charged anyons or upstream charge-neutral anyons, and they are not further fractionalized while flowing." In the section of "Non-equilibrium correlator of anyon collider".

Reviewer's comment-2: "I also have a related questions, which is even more puzzling for me, regarding the differential conductance, eq. (9). As far as I understand, the main text discusses the case where the charge mode propagates towards QPCc, in which case I understand how to measure the tunneling current resulting from the injection of currents I_A and I_B . However, I do not understand how one can define the differential conductance in the other regime where the neutral mode is propagating towards QPCc. It seems to me that the tunneling current is always zero as no net current propagates towards QPCc (currents I_A and I_B propagate towards DA and DB on Fig S3). I would thus conclude that, in this case, I_T is always zero and independent of I_A and V_{ref} and I do not understand how $Perf$ is defined in this case?"

Response: We thank Reviewer 2 for this comment. At first sight, it would be surprising how the injection of only the neutral non-Abelian anyons induces I_T . As discussed above, tunneling of a neutral anyon is always accompanied by tunneling of an Abelian charged anyon. Hence, I_T can be nonzero in general when the neutral non-Abelian anyons are injected to QPC_C. Nevertheless, for the states of our interest such as anti-Pfaffian, particle-hole Pfaffian, and anti-Read-Rezayi, the leading term of I_T in Eq. (2) vanishes. This is not because the injected anyons are charge neutral. It is because the imaginary part of the monodromy is zero for the non-Abelian anyons; when the imaginary part of the monodromy is zero, the braiding process induced by anyon tunneling from Edge A to Edge B at QPC_C destructively interferes with the braiding by anyon tunneling from Edge B to Edge A, making I_T be zero, as shown in Eqs. (1) and (2). This is why P_- of Eq. (4) diverges. When V_{ref} is applied, a small voltage bias is induced across QPC_C and I_T can have a finite value. The finite value of I_T depends on the amount of the injection of the non-Abelian anyons, and has the information of the braiding and the monodromy.

In the revised manuscript, we explicitly point out Eqs. (1) and (2) in the discussion of the destructive interference shown in I_T by the sentence "As a notable result, the time-domain interference contributes to the current I_T destructively as shown in Eqs. (1) and (2)."

Reviewer's comment-3: "Finally, Fig 2 looks like a useful of the monodromy but I find it hard to connect it to the Fig1.c, that is, to understand how the diagram in Fig.2 enters in the description of the collision experiment. Is it possible to make this connection more explicit?"

Response: We thank the reviewer for the comment. This is also useful for improving the readability of our manuscript. To explicitly connect the figure (which is now Fig. 3 in the revised manuscript, as a new figure is added as Fig. 2) with Fig. 1, we add the following in the figure caption of: "The red and blue loops correspond to anyons that tunnel at QPC_A/B and QPC_C respectively [See the loops of the same colors in Fig. 1(c)]. Untying the topological link between the loops amounts to the monodromy M (or M^* depending on the direction of the loops)."

Thanks to the comments, we improve our manuscript. We would like to thank Reviewer 2 very much.

REVIEWERS' COMMENTS

Reviewer #1 (Remarks to the Author):

The authors have made satisfactory changes. I recommend publication of the manuscript in its present form.

Reviewer #2 (Remarks to the Author):

I am fully satisfied with the author's answers to the referee's questions and comments. I am also satisfied with the revisions on the manuscript. The paper can be published in Nature Communications.